# 6-Shogaol Exhibits a Promoting Effect with Tax via Binding HSP60 in Non-Small-Cell Lung Cancer

**DOI:** 10.3390/cells11223678

**Published:** 2022-11-19

**Authors:** Shulipan Mulati, Rongsong Jiang, Jinfeng Wang, Yicun Tao, Weiyi Zhang

**Affiliations:** School of Pharmacy, Xinjiang Medical University, Urumchi 830017, China

**Keywords:** 6-Shogaol, non-small-cell lung cancer, HSP60, proteasome, toxicity

## Abstract

Non-small-cell lung cancer (NSCLC) is a prevalent malignant tumor with high morbidity and mortality rates worldwide. Although surgical resection, adjuvant radiotherapy/chemotherapy, and targeted molecular therapy are the cornerstones of NSCLC treatment, NSCLC is associated with high recurrence rates and drug resistance. This study analyzed the potential targets and pathways of 6-Shogaol (6-SH) in NSCLC, showing that 6-SH binds to heat-shock 60 kDa protein (HSP60) in A549 cells, induces cell apoptosis, and arrests the cell cycle possibly by disrupting the mitochondrial function. HSP60 was identified as the target of 6-SH and 6-SH-induced HSP60 degradation which was mediated by the proteasome. The binding of 6-SH with HSP60 altered its stability, inhibited the ERK, Stat3, PI3K, Akt, and mTOR signaling pathways, and Tax acted synergistically with 6-SH, indicating that 6-SH could be developed as a potential therapeutic agent for an NSCLC treatment.

## 1. Introduction

An estimated 2 million new cases and 1.76 million deaths occur annually from lung cancer, ranking it the leading cause of worldwide cancer-related deaths [1] and mortality [2]. Like most malignancies, it is caused by several etiologies, and it is categorized into non-small-cell lung cancer (NSCLC, approximately for 85% of all diagnoses) and small-cell lung cancer (approximately 15% of all diagnoses) [3]. The most frequent subtype of lung cancer is adenocarcinoma, which is followed by squamous-cell carcinomas in NSCLC [4,5]. Early-stage NSCLC patients frequently undergo surgery, whereas molecular targeted therapy is commonly used for individuals with advanced NSCLC [6]. Despite significant advances being made in the early identification and treatment options in recent years, the 5-year survival rate has remained at 20% [7,8]. Consequently, further research on the pathophysiology and molecular mechanisms of NSCLC is required to identify the novel biomarkers for its early diagnosis and the prediction of the treatment outcomes and prognosis and to identify novel therapeutic targets.

6-Shogaol (6-SH) is one of the most significant bioactive compounds that is found in dried ginger [9], with it having highly potent antioxidant, anti-inflammatory, antiemetic, and other pharmacological effects due to the conjugation of its α,β-unsaturated ketone skeleton [10]. Recently, 6-SH has been reported to exert anti-tumor properties in cervical cancer, leukemia, oral squamous carcinoma, etc. [11,12,13,14], and it can suppress the proliferation of NSCLC cells [15], but the mechanism and target proteins remain unknown.

Heat-shock proteins (HSPs) play a crucial role in maintaining cellular homeostasis as molecular chaperones [16], and they are upregulated in malignant tumor cells [17]. Therefore, the HSP expression levels can predict the occurrence of various cancers [18]. Moreover, HSP60 affects the occurrence, development, metastasis, and other processes of the entire tumor, and it is highly expressed in various tumor cells, including breast cancer, ovarian cancer, and NSCLC [19,20,21].

The present study evaluated the effects of 6-SH in NSCLC, showing that 6-SH can cause cell cycle arrest and apoptosis in NSCLC cell lines. Based on virtual screening using network pharmacology, 6-SH has been indicated to be a promising drug target for the treating and preventing NSCLC. Furthermore, the binding of 6-SH to HSP60 changed its stability, inhibited the ERK, Stat3, PI3K, Akt, and mTOR signaling pathways, and acted synergistically with Tax. These results suggest that 6-SH could be developed as a potential therapeutic agent for an NSCLC treatment.

## 2. Materials and Methods

### 2.1. Reagents and Antibodies

The antibodies against Bax (50599-2-lg), Bcl-2 (60178-1-lg), and survivin (10988-1-AP) were obtained from Proteintech (Chicago, IL, USA). The anti-GADPH (bsm-0978M), HSP60 (bs-0191R) and β-actin (bs-0061R) antibodies were obtained from Bioss (Beijing, China). The AKT (# 9272S), P-AKT (# 86758S), PI3K (# 4255S), P-PI3K (# 17366S), mTOR (# 2983S), P-mTOR (# 5536S) antibodies were purchased from CST (BCG, USA). 6-SH (SS8510), Taxol (Tax, SP8020) and D, L-dithiothreitol (DTT, A285) were purchased from Solarbio (Beijing, China). 3-azido-7-hydroxycoumarin (M0135–73235) was obtained from Macklin (Shanghai, China).

### 2.2. Cell Culture and Cell Viability Assessment

All of the different kinds of cells and the culture media were provided by the Procell (Wuhan, China). Among them, the A375, A549, H8, U251, 3T3-L1 and PANC-1 cells were cultured in DMEM. The TSSCA and HK-2 cells were cultured in RPMI-1640 medium. All of the cells were supplemented with 10% fetal bovine serum, 100 units/mL penicillin, and streptomycin at 37 °C in 5% CO_2_. The cytotoxicity of 6-SH and the other medicines was determined using the Cell Counting Kit-8 (CCK-8, bs-0764P, Bioss, Beijing, China). Additionally, the CompuSyn software was used to calculate the cooperativity index (CI) value, which reflects the synergistic effects of the two medicines.

### 2.3. Colony Formation Assays

The A549 cells were seeded into six-well plates (800 cells per well) containing 6-SH (0–20 μM) and incubated for 12 days. Then, the cells were fixed in methanol for 15 min and stained with Giemsa for 30 min, after which the colonies were photographed (Olympus GX41, Tokyo, Japan).

### 2.4. Cell Migration Assays

Once the A549 cells had reached 90% confluence, the monolayers were scratched using disposable yellow tips and washed in phosphate-buffered saline (PBS). The cell migration was monitored by photographing each wound after 48 h (Olympus GX41, Tokyo, Japan). The migration inhibition rate was calculated as follows: inhibition rate (%) = [1 − (average width in the drug-treated group/average width in the ctrl group)] × 100%

### 2.5. Xenografts In Vivo

The 6-week-old female BALB/c-nude mice (16–18 g, specific pathogen-free, SPF) were obtained from the Laboratory Animal Experimental Center of the Academy of Military Medical Sciences (SCXK2012-0004, Beijing, China), and they were injected with A549 cells (1 × 10^7^). Two weeks after the tumor formation, the mice were treated with Tax (5 mg/kg/day) and 6-SH (5, 10, or 20 mg/kg/day) for one week. The negative controls were saline. The tumor volume was measured every three days and calculated as length × width^2^/2. All of the protocols related to the animal care and use have been approved (IACUC-20202304-09).

### 2.6. Cell Cycle Analysis

The A549 cells were incubated with 6-SH (0–40 µM) for 24 h before washing them with PBS and fixed in ice-cold (75% (*v*/*v*)) ethanol overnight at −20 °C. The DNA of the resuspended cells was analyzed using a cell cycle kit (CA1510, Solarbio, Beijing, China) and a flow cytometer (BD LSR Fortessa, NJ, USA).

### 2.7. Hoechst 33342 Staining Assay

6-SH (0–80 µM) was added to A549 cells for 24 h before the cells were washed in PBS and stained with Hoechst 33342 (a blue, fluorescent dye which can penetrate cell membranes; when the apoptosis occurred, the nuclei of apoptotic cells were stained). The apoptotic cells were visualized using an inverted fluorescence microscope (Leica TCS SP8, Japan).

### 2.8. Annexin V-FITC/PI Double-Staining Assays

Following a treatment with 0–80 µM 6-SH for 24 h and washing them with PBS, the apoptosis was detected using an Annexin-V-FITC detection kit (A005-3, Solarbio, Beijing, China) and a flow cytometer (BD LSR Fortessa, NJ, USA).

### 2.9. Measurement of Mitochondrial Membrane Potential (MMP)

After incubating them with 0–80 µM 6-SH for 24 h, the A549 cells were stained with JC-1 for 20 min according to the manufacturer’s instructions (M8650, Solarbio, Beijing, China). CCCP was used as a positive control. Finally, the MMP was measured using a flow cytometer (BD LSR Fortessa, NJ, USA).

### 2.10. Network Pharmacology Analysis

The three-dimensional (3D) structure of 6-SH was determined using ChemBio3D Ultra 14.0 software, and then, it was uploaded into Pharm Mapper Database for reverse docking to predict the putative targets. The candidate targets were then inputted into the KEGG analysis and String 11.0 database for a protein functional enrichment analysis and a PPI network construction.

### 2.11. Enrichment of Target and Western Blotting Assay

An AL-6-SH probe was synthesized and used to create the functionalized AL-6-SH magnetic microspheres (AL-6-SH-MMs) for target fishing. The A549 cells were cultured using an Al-6-SH probe for 6 h before being subjected to SDS-PAGE, Western blotting, and HPLC-mass spectrometry to identify the captured protein target (Huada Gene Research Center, Beijing, China).

### 2.12. Cellular Thermal Shift Assay (CETSA)

The A549 cell lysates were treated with 6-SH (20 μM) for 24 h at different temperatures, and the HSP60 protein levels were quantified by Western blotting.

### 2.13. Bioinformatic Analysis

The HSP60 pan-cancer expression profile was analyzed using the TIMER. Available online: http://timer.cistrome.org/ (accessed on 3 May 2022). The TCGA. Available online: https://portal.gdc.cancer.gov (accessed on 3 May 2022) was used to analyze the relationship between the HSP60 expression and the prognosis of lung adenocarcinoma (LUAD) and the lung squamous carcinoma (LUSC). The difference in the survival rate was compared using a log-rank test, with the predictive value of HSP60 mRNA which was evaluated by a timeROC analysis. Kaplan–Meier curves were used to calculate the *p*-values and hazard ratios (HR) with a 95% confidence interval (CI). In all of the analyses, R (version 4.0.3, Available online https://www.r-project.org (accessed on 20 October 2020), which was developed by Ross Ihaka and Robert Gentleman of Auckland University (Auckland, New Zealand)) was used. A Spearman’s correlation analysis was performed to determine the correlations between the non-normally distributed quantitative variables.

### 2.14. Molecular Dynamics (MD) Assay

The three-dimensional structure of HSP60 (PDB ID: 6mrc) was obtained from PDB database for molecular docking via AutoDock Vina 1.1.2 and MD analysis using AMBER 18. The molecular mechanics/Generalized Born Surface Area (MM/GBSA) calculations were then used to calculate the binding free energies between the proteins and ligands in all of the systems.

### 2.15. Developmental Toxicity Test of Zebrafish Embryos

The AB-wild type adult *zebrafish* (3 months old) were purchased from the Institute of Aquatic Biology, Chinese Academy of Sciences (Wuhan, China). During the breeding stage, the male and female fish were placed in separate spawning boxes at a 1:1 ratio before mixing them in the early morning to lay eggs. During the exposure period, 96-well plates containing normal embryos were randomly selected and viewed daily under a light microscope to assess the developmental toxicity.

### 2.16. Statistical Analysis

The results are presented as mean ± standard deviation (SD), with t tests being used to analyze the significant differences between the two groups, and an analysis of variance was used for multiple groups. A *p*-value < 0.05 was considered to be statistically significant.

## 3. Results

### 3.1. 6-SH Inhibited A549 Cells Growth In Vitro and In Vivo

To test the specificity of 6-SH, multiple types of cancerous and normal cells were identified. 6-SH inhibited the A549 cells more effectively than it did for the other cancer cells, and it was found less hazardous to normal cell lines (Figure 1B) so the subsequent experiments were conducted using the A549 cells. The 50% growth inhibition concentration (IC_50_) values of 6-SH against the A549 cells were 48.67, 77.33, and 111.33 μM after 24, 48, and 72 h, respectively, thus displaying concentration- and time-dependent patterns (Figure 1C). Morphological modifications in the cells were detected at various drug concentrations (Figure 1D). Furthermore, the colony-forming test and migration assays established that 6-SH inhibits cell proliferation. Compared to the “Ctrl” group, 6-SH significantly reduced the number of colonies and impaired the cell migration in a dose-dependent manner (Figure 1E,F). 6-SH also dose-dependently inhibited the tumor growth in the xenograft model (Figure 1G–I) in comparison to that in the control group. Taken together, these findings suggest that 6-SH substantially inhibits A549 cell growth in vitro and in vivo.

### 3.2. 6-SH Induced Apoptosis and Arrested the G0/G1 Phase in A549 Cells

To detect whether the cell cycle arrest contributed to the 6-SH-induced cell growth inhibition, the cell cycle distribution was analyzed by flow cytometry. As displayed in Figure 2A, 6-SH markedly induced the G0/G1 phase arrest in the A549 cells, which was accompanied by an increase in the percentage of apoptotic cells from 6.4% to 21.4% (Figure 2B) and increased nuclear chromatin blue fluorescence (Figure 3C). The 6-SH treatment reduced the expression of anti-apoptotic proteins such as Bcl-2 and survivin and increased the pro-apoptotic protein Bax (Figure 2D,E and Appendix A). Furthermore, the increasing concentration of 6-SH disrupted the MMP (Figure 2F). These results indicated that both the G0/G1 phase arrest and the cell apoptosis triggered by 6-SH contributed to the cytotoxicity through the mitochondrial pathway in the A549 cells.

MMP is frequently damaged during the apoptotic process, and JC-1 can be utilized to indicate the mitochondrial potential in multiple cell types. As a result, we labeled the cells using a JC-1 probe and used fluorescence confocal microscopy to examine the changes in the MMP following the 6-SH treatment. The results revealed that as the 6-SH concentration increases, the green fluorescence considerably increases, indicating that 6-SH may have destroyed the MMP (Figure 2F). The above results indicated that both the G0/G1 phase arrest and the cell apoptosis triggered by 6-SH contributed to the cytotoxicity through mitochondrial pathway in the A549 cells.

### 3.3. Network Pharmacology Analyzed the Potential Targets and Pathways of 6-SH Anti-NSCLC

To identify the target proteins of 3D 6-SH structure which were involved in the anti-NSCLC treatment, the Pharm Mapper Database was first applied. The top 40 protein targets were selected according to the relative enrichment scores (Figure 3A). A KEGG analysis to elucidate the 6-SH target proteins and related pathways revealed the involvement of multiple signaling pathways, such as the mTOR, JAK/STAT, and PI3K/Akt/mTOR pathways (Figure 3B). Meanwhile, a GO analysis showed that the abnormal activation of these systems can lead to cell malignancy and tumor development (Figure 3C).

### 3.4. 6-SH Targeted HSP60 and Reduced Its Protein Stability

A 6-SH probe and AL-6-SH-MMs were prepared to detect the protein targets in the A549 cells (Figure 4A). One distinct band was observed (Figure 4B, lane 3, about 60 kDa), so the captured protein was enzymatically hydrolyzed and identified by HPLC-MS/MS as HSP60 with a coverage rate of 69% (Figure 4C). The Western blotting verified that the captured protein was HSP60 (Figure 4B, lane 2), and CETSA revealed that the binding changed the protein stability of HSP60 in a temperature-dependent manner (Figure 4D and Appendix A).

The A549 cells were exposed to nonactin, an inhibitor of HSP60, to further verify that HSP60 is the target of 6-SH. The A549 cells became dependently less sensitive to the 6-SH concentration after the nonactin pretreatment (Figure 4E,F and Appendix A), confirming that HSP60 was the key target protein in the 6-SH-induced apoptosis. Moreover, the A549 cells were pre-incubated with MG-132 (a proteasome inhibitor) before the 6-SH treatment to determine whether the proteasome-mediated degradation of HSP60 was also triggered by 6-SH, demonstrating that 6-SH, alone, reduced HSP60, which was abrogated by MG-132, indicating that 6-SH-induced HSP60 degradation is a proteasome-mediated process (Figure 4G). Hence, the above findings suggest that HSP60 is a potential target of 6-SH in NSCLC.

### 3.5. Clinical Relevance of HSP60 in Lung Cancer

To further analyze the HSP60 expression in cancer, we explored the significance of the HSP60 expression in patients using the public TCGA database. The HSP60 mRNA expression was significantly higher in 16 tumors, including LUAD and LUSC, when it was compared to the adjacent tissues, suggesting that HSP60 may be a biomarker for predicting a poor prognosis (Figure 5A). As depicted in Figure 5B,C, the LUAD patients with higher HSP60 mRNA expressions had a worse OS and prognosis when they were compared to the patients with a lower HSP60 expression. On the contrary, the LUSC patients had a slightly better OS and prognosis when they were compared to the low HSP60 expression group. The tumor mutational burden (TMB) is an emerging biomarker that is related to immunotherapy, thus, we investigated the relationship between the HSP60 expression and TMB, showing that HSP60 expression in LUAD and LUSC positively correlated with the TMB score (Figure 5D,E). The findings support our pre-clinical findings that HSP60 contributes to lung cancer progression.

### 3.6. 6-SH Down-Regulated ERK, Stat3, PI3K, Akt, and mTOR, Combined with Tax to Promote Anti-NSCLC Effects

Additionally, we validated the most correlated pathways that were predicted by the network pharmacology (Figure 3). The results showed that 6-SH suppressed the phosphorylation of ERK, Stat3, PI3K, Akt, and mTOR, but it did not affect their total protein levels (Figure 6A–F and Appendix A). Tax is a commonly used chemotherapy drug, but it is associated with side effects. To detect the promoting effect of 6-SH and Tax, the A549 cells were incubated with 6-SH alone or in combination with Tax (the tested concentrations were close to its IC_50_, Figure 6G), demonstrating that Tax acted synergistically to enhance the inhibitory effect of 6-SH (Figure 6H–J).

### 3.7. Hydrophobicity Was the Main Binding Mode of 6-SH to HSP60

The Amber 18 program was used to verify the potential position of 6-SH binding on HSP60. The root-mean-square deviations (RMSD) were plotted based on the starting structure along the simulation time (Figure 7A) to reflect the protein flexibility since the protein flexibility can be reduced after the drug binds to the protein, thus stabilizing the protein. As illustrated in Figure 7B, the RMSF of the 6-SH binding site (highlighted in red) was within 2 Å, suggesting that 6-SH may stabilize the active site. Moreover, the top ten amino acids contributing to the binding of 6-SH and HSP60 (ILE 150, PRO 33, MET 482, SER 151, GLY 53, GLY 154, ASN 153, GLY 32, THR 90, and THR 149), and ILE 105 and PRO 33 are hydrophobic (binding free energy < −1.5 kcal/mol), suggesting that hydrophobicity may be more critical in this binding process (Figure 7C). Furthermore, 6-SH formed hydrogen bonds with GLY 53 of HSP60, and as well as this, it had a hydrophobic interaction with THR 90, LYS 51, ILE 150, and PRO 33 (Figure 7D), indicating that hydrophobicity and hydrogen bonding are the main binding forces.

### 3.8. Effects of 6-SH on Toxicity in Zebrafish Embryos

*Zebrafish* are used widely as aquatic models in toxicology research, and the toxicity of 6-SH was evaluated in the early developmental stages of the embryos. As demonstrated in Figure 8A,B and Table 1, the hatching rates significantly decreased at 24–96 h after the 200 and 400 μM 6-SH treatment. A significant increase in the mortality rates was observed after 96 h with the higher concentrations.

## 4. Discussion

NSCLC is a cancer type that is prone to relapse and metastasis [22]. The discovery of molecular-targeted tumor therapy, which targets tumor cells specifically, can reverse cancerous cells at the molecular level, offering new possibilities for anti-cancer therapy [23]. However, acquired drug resistance is becoming an increasingly widespread issue in clinical practice [24], thus, identifying novel medicines and procedures is critical for improving the patient outcomes. One of the underlying processes of NSCLC is the over-expression of the anti-apoptotic protein Bcl-2 and the low expression of the apoptosis-promoting protein Bax, which slows down cell death and extends the cell survival [25,26]. The current study found that 6-SH may activate Bax expression, suppress Bcl-2 expression, and disrupt the MMP in A549 cells, indicating that 6-SH can trigger the mitochondria-mediated apoptosis pathway and induce cell apoptosis.

Network pharmacology is a research approach that is based on systems biology and multi-directional pharmacology. The common targets of the medications and diseases have been investigated as well as the biological activities and signaling pathways using a drug–target–disease multi-level interaction network [27]. The association between 6-SH and NSCLC was proven using a PPI network and a KEGG biological pathway enrichment analysis to further investigate the mechanism underlying the anti-cancer effect of 6-SH in NSCLC. The ERK, Stat3, PI3K, Akt, and mTOR signaling pathways were associated with 6-SH anti-NSCLC activity.

According to classical genetics, a vital physiological function is usually controlled by several genes, and small-molecule medicines affect the genes in a concentration-dependent manner. Therefore, a synergistic molecule can markedly enhance the synergistic or antagonistic effects [28]. HSP60 is a molecular chaperone that is mainly located in the mitochondrial matrix [29], and it is a potential biomarker for tumor diagnosis and prognosis, participating in cell death and cell cycle regulation and affecting the survival and metastasis of various tumors, as well as mediating drug resistance [30,31]. For example, inhibiting HSP60 expression can reverse drug resistance in 5-FU-resistant colorectal cancer cells [32]. The present study demonstrated that Tax acted synergistically to enhance the effect of 6-SH.

Although HSP60 is over-expressed in NSCLC when it is compared to non-cancerous lung tissues and it may be a novel biomarker of poor prognosis for NSCLC patients [33], its biological role remains unclear. A pan-cancer expression analysis and a prognostic analysis revealed that patients with a high HSP60 expression level had a worse prognosis than the low-HSP60-expression LUAD patients. Furthermore, HSP60 expression is positively correlated with the TMB score in LUAD and LUSC patients. Taken together, this clinical analysis demonstrates that HSP60 may be a biomarker for predicting a poor prognosis in lung cancer.

HSP60 is a survivin-related protein, and it can form an HSP60-survivin complex in the mitochondria and regulate the stability of survivin, thereby maintaining its normal anti-apoptotic function [34]. Our results revealed that 6-SH could inhibit the survivin expression, disrupt the MMP in the A549 cells and eventually induce cell apoptosis by binding to HSP60. The binding energy of 6-SH and the HSP60 protein was −22.39 ± 1.68 kcal/mol, indicating that they have a strong binding force, with hydrophobicity being the main binding mode of 6-SH to HSP60. To the best of our knowledge, this is the first study to substantiate that 6-SH targets HSP60 to play an anti-NSCLC role.

As an animal model for drug toxicity testing, *zebrafish* embryos are commonly used. It is easy to use *zebrafish* embryos as they are optically transparent and they are highly sensitive to toxicants. The drug toxicity analysis revealed that high concentrations of 6-SH induced some toxicity symptoms, including an increased mortality and hatching inhibition, but 6-SH is safe at normal concentrations, and it is therefore, a potential anti-cancer drug candidate.

## 5. Conclusions

This study demonstrated that 6-SH could arrest the cell cycle and induce cell apoptosis, possibly via regulating Bcl-2 family and mitochondria. 6-SH bound to HSP60 and inhibited the ERK, Stat3, PI3K, Akt, and mTOR pathways, and 6-SH combined with Tax had a synergistic effect. These findings provide a new understanding and indicate the potential of 6-SH for the NSCLC treatment.

## Figures and Tables

**Figure 1 cells-11-03678-f001:**
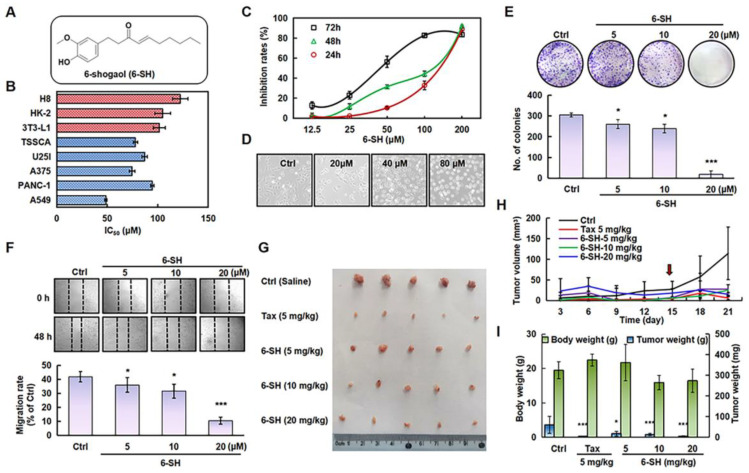
6-SH inhibited the growth of A549 cells in vitro and in vivo. (**A**) Structure of 6-SH. (**B**) The different cell lines were treated with 0–200 µM 6-SH for 72 h, the IC_50_ value was measured (*n* = 3). (**C**) Cells were treated with 0–200 µM 6-SH for different hours, and the cell viability was assayed (*n* = 3). (**D**) The effect on cell density of 6-SH (0–80 μM) for 24 h. (**E**) Cells were incubated with 0–20 µM 6-SH for 14 days, the Giemsa staining was performed; histogram shows the number of colonies (*n* = 3). (**F**) Cells were incubated with 0–20 µM 6-SH for 24 h, the cell migrations were analyzed, and the histogram shows the migration rate (*n* = 3). (**G**–**I**) A549 cells were injected subcutaneously into nude mice. Then, they were treated with Tax (5 mg/kg/day, as a positive control) and 6-SH (5, 10, or 20 mg/kg/day) for one week after tumor formation (the red arrow at Day 14 represents the first day of treatment). The tumor volume was calculated (*n* = 5). Values represent the means ± SD. * *p* < 0.05, *** *p* < 0.001 versus “Ctrl” group.

**Figure 2 cells-11-03678-f002:**
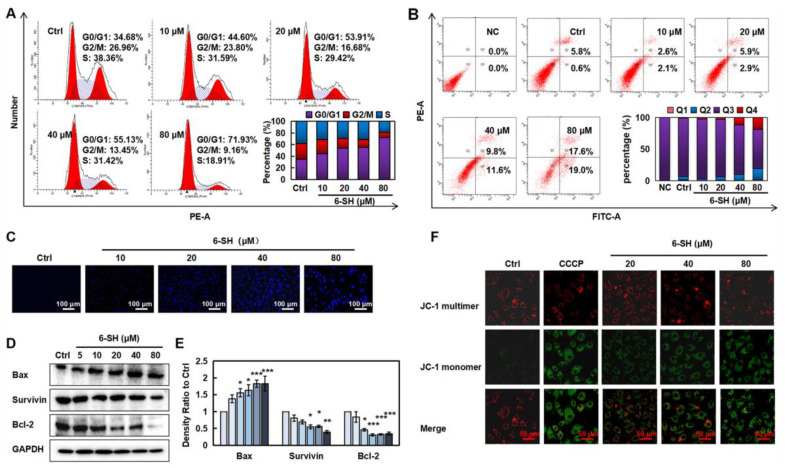
6-SH arrested the cell cycle and induced apoptosis. (**A**) Cells were cultured with 0–80 µM 6-SH for 24 h, then, the DNA content was texted. Histograms show the percentage of cells in different phase. (**B**) Cells were treated with 0–80 µM 6-SH for 24 h. 6-SH-induced apoptosis was detected. “Q2” and “Q4” represent late and early apoptosis in histograms, respectively. (**C**) Cells were treated with 0–80 µM 6-SH for 24 h, and the changes of nuclear chromatins were detected. (**D**,**E**) Cells were treated with 0–80 µM 6-SH for 24 h, and the indicated proteins were assayed. Histograms show the density ratio (*n* = 3). (**F**) Cells were treated with 0–80 µM 6-SH for 24 h, and the MMP was analyzed using JC-1 probe. Values represent the means ± SD. * *p* < 0.05, ** *p* < 0.01, *** *p* < 0.001 versus “Ctrl” group.

**Figure 3 cells-11-03678-f003:**
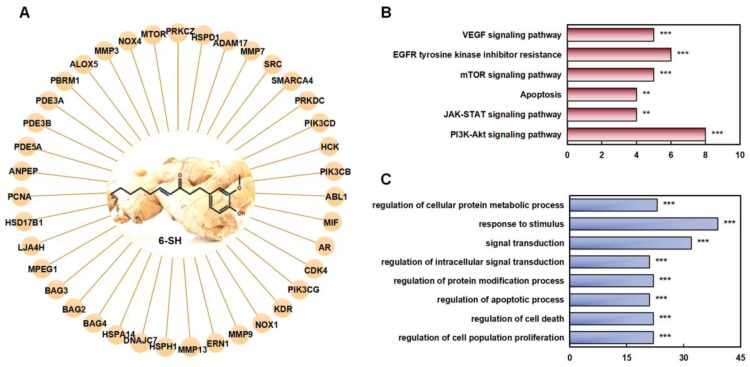
Prediction of the targets of 6-SH. (**A**) The function assays for 6-SH with PPI network. (**B**) KEGG analysis. (**C**) GO analysis, ** *p* < 0.01, *** *p* < 0.001 was considered statistically significant.

**Figure 4 cells-11-03678-f004:**
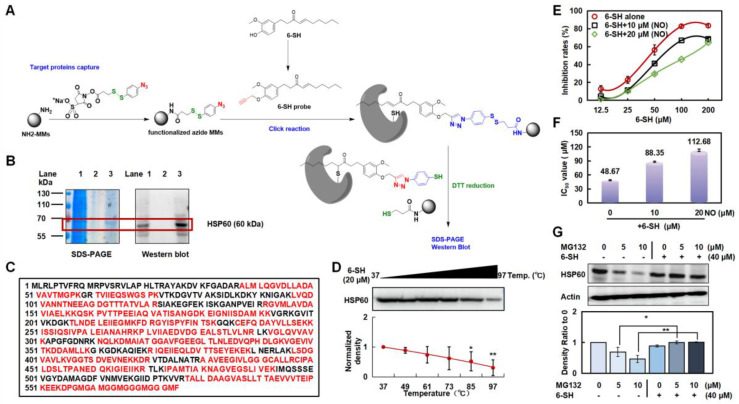
6-SH targeted HSP60 and inhibited its stability. (**A**) The modification process of AL-6-SH-MMs and the technology roadmap for target fishing method. (**B**) SDS-PAGE and Western blotting analysis of captured protein by 6-SH probe (Lane 1: A549 lysate; Lane 2: a negative control without the 6-SH probe; Lane 3: 6-SH probe-captured proteins). (**C**) The sequence of HSP60 protein and the identified peptide (red). (**D**,**E**) 6-SH treatment (20 µM) inhibited the thermal stability of HSP60 measured by CETSA; the statistics show the normalized density (*n* = 3). Values represent the means ± SD. * *p* < 0.05, ** *p* < 0.01 versus “Ctrl” group. (**E**,**F**) Cells were pre-treated with nonactin (10 or 20 μM) for 4 h, and then, they were treated with 6-SH for 72 h. The inhibition and IC_50_ value of 6-SH were assayed. (**G**) A549 cells were treated with MG-132 (5 or 10 μM) for 1 h before the addition of 40 μM 6-SH for another 5 h, and then, the HSP60 was examined; the statistics show the normalized density (*n* = 3). Values represent the means ± SD. * *p* < 0.05, ** *p* < 0.01 versus corresponding group.

**Figure 5 cells-11-03678-f005:**
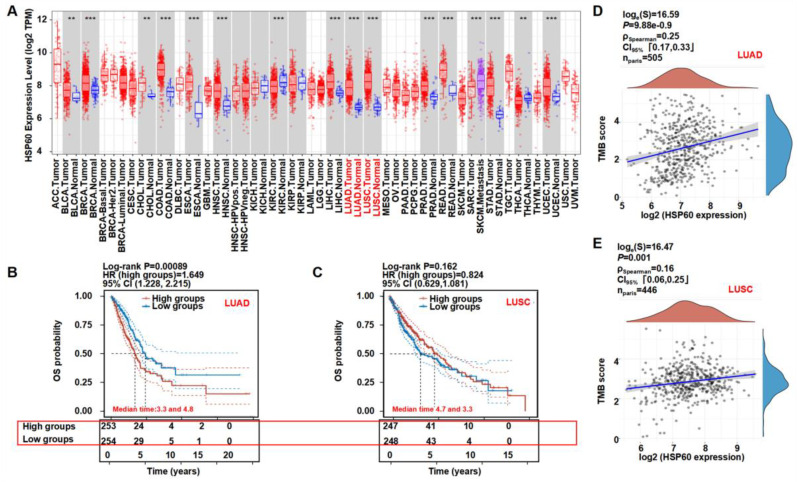
Clinical relevance of HSP60 in lung cancer. (**A**) Diffuse expression analysis of HSP60, ** *p* < 0.01, *** *p* < 0.001. (**B**,**C**) Prognostic analysis of HSP60 in LUAD and LUSC. (**D**,**E**) Spearman correlation analysis of TMB and HSP60 expression in LUAD and LUSC. *p* < 0.05 was considered to be statistically significant.

**Figure 6 cells-11-03678-f006:**
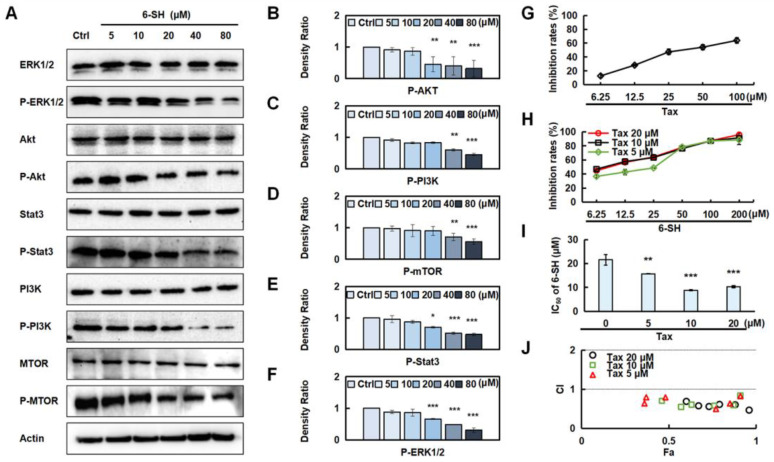
Validation of 6-SH anti-NSCLC pathway. (**A**) Cells were treated with 0–80 µM 6-SH for 24 h, and the indicated proteins were assayed. (**B**–**F**) Histograms show the density ratio (*n* = 3). Values represent the means ± SD. * *p* < 0.05, ** *p* < 0.01, *** *p* < 0.001 versus “Ctrl” group. (**G**–**I**) Cells were treated with Tax or Tax combined with 6-SH for 72 h, respectively, the inhibition rate and IC_50_ were assayed. (**J**) The CI value was calculated. CI values of <1 indicate synergistic effect. Values represent the means ± SD. * *p* < 0.05, ** *p* < 0.01, *** *p* < 0.001 versus “0” group.

**Figure 7 cells-11-03678-f007:**
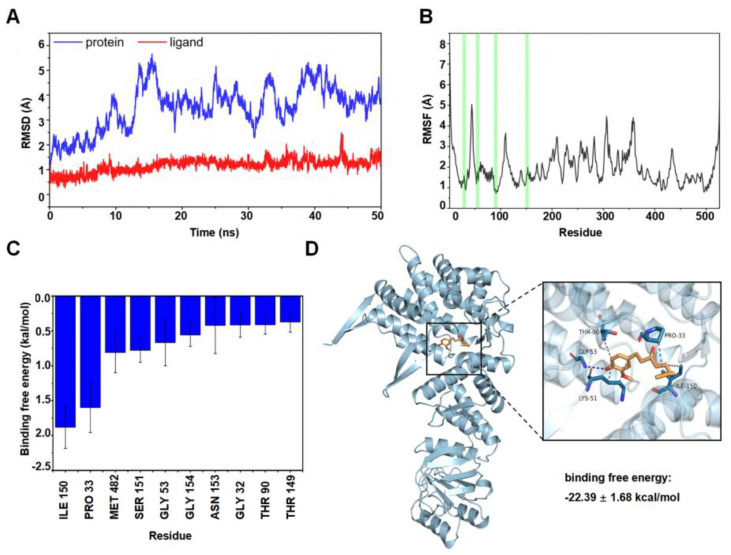
MD analysis. (**A**) RMSD of the complex over time. (**B**) RMSF was calculated, and the green line represents the location of the amino acids of the small molecule binding site. (**C**) Analysis of binding free energy. (**D**) Molecular docking analysis of HSP60 and 6-SH (blue: HSP60, orange: 6-SH).

**Figure 8 cells-11-03678-f008:**
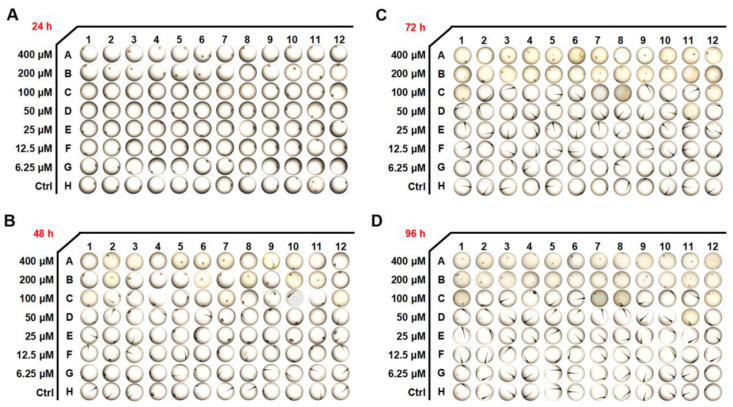
Developmental effects of 6-SH on embryos. (**A**–**D**) The embryo morphology after 6-SH treatment for 24–96 h (*n* = 12).

**Table 1 cells-11-03678-t001:** The mortalities and abnormality rates after 6-SH treatment for 96 h.

6-SH (μM)	Survival Number	Number of Deaths	Number of Deformities	Mortalities	Abnormality Rates
400	0	12	0	100.00%	0.00%
200	0	12	1	100.00%	0.00%
100	7	5	0	41.67%	0.00%
50	11	1	0	8.33%	0.00%
25	12	0	0	0.00%	0.00%
12.5	12	0	0	0.00%	0.00%
6.25	12	0	0	0.00%	0.00%
Ctrl	12	0	0	0.00%	0.00%

## Data Availability

All data generated or analyzed during this study are included in this published article and its Appendix A.

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
