# Peer review of "6-Shogaol Exhibits a Promoting Effect with Tax via Binding HSP60 in Non-Small-Cell Lung Cancer"

_cells, 2022, doi:10.3390/cells11223678_

Round 1

Reviewer 1 Report

The study aims to analyze the potential targets and pathways of 6-Shogaol in NSCLC. The study seems interesting and novel however the method section must be written clearly. Several methods information was missing. Some of the major drawbacks are:

-        It was mentioned in the text that 6-SH acted synergistically with Tax. Although Figure 6 shows the CI and other parameters of the synergism, it is impossible to compare the dose effects as there is no explanation bar in figure 6 B-I.

-        Section 2.3. can be written clearly.

-        Photographing information can be added in given sections.

-        Tumor volume measurement formula must be corrected.

-        The confocal microscopy information must be added to section 2.6, 2.7 and 2.8 (according to line 189).

Author Response

We are very grateful to you for giving us an opportunity to revise our manuscript. We want to thank the editors and reviewers for reviewing of manuscript and providing constructive suggestions on our manuscript entitled “6-Shogaol exhibits a promoting effect with Tax via binding HSP60 in non-small cell lung cancer” (Manuscript ID: cells-1994604).

We have studied reviewers’ comments carefully and tried our best to revise our manuscript according to the comments. Thanks again to the hard work! Below are the detailed responses:

Reviewer 1

The study aims to analyze the potential targets and pathways of 6-Shogaol in NSCLC. The study seems interesting and novel however the method section must be written clearly. Several methods information was missing. Some of the major drawbacks are:

Comments 1: It was mentioned in the text that 6-SH acted synergistically with Tax. Although Figure 6 shows the CI and other parameters of the synergism, it is impossible to compare the dose effects as there is no explanation bar in figure 6 B-I.

Response: We thank the reviewer for this suggestion. As requested, we have added related information and replaced figure in manuscript (result 3.6, lane 272).

Comment 2: Section 2.3. can be written clearly.

Response: We accepted the reviewer’s suggestion and added related information (lane 77).

Comment 3: Photographing information can be added in given sections.

Response: We are very grateful to reviewer for suggesting this question. As you say, we have added the related information (methods 2.3 and 2.4, lane 77 and 82) and marked it in red.

Comment 4: Tumor volume measurement formula must be corrected.

Response: As the reviewer suggested, we have corrected the tumor volume measurement formula (method 2.5, lane 91) and marked it in red.

Comment 5: The confocal microscopy information must be added to section 2.6, 2.7 and 2.8 (according to line 189).

Response: We thank the reviewer for the suggestions, and we have added related information in manuscript (methods 2.6, 2.7 2.8 and 2.9, lane 97, 102, 107 and 112) and marked it in red.

Reviewer 2 Report

This study analyzed the potential targets and pathways of 6-Shogaol (6-SH) in NSCLC and demonstrated that 6-SH could be developed as a potential therapeutic agent for NSCLC treatment. The authors demonstrated 6-SH inhibited cancer cell growth both in vitro and in vivo, through induction of apoptosis and cell cycle arrest. Direct target of 6-SH was identified and verified, and other predicted signaling pathways was also validated.

These results have important clinical significance for the treatment of tumors. However, some issues need to be addressed before it could be further considered.

Minor issues:

1.      In Figure 2C, increased nuclear chromatin staining reflects increased chromatin condensation, which is a morphologic hallmark of apoptosis. Please explain more in the result section for the readers who are not that familiar with this feature.

Although Hoechst staining was mentioned in the method section, it should still be briefly mentioned either in the main text, in the figure caption, or labeled in Figure 2C.

2.      Figure 3A was not cited in the result section 3.3.

3.      In Figure 4A, it seems with 6-SH probes were first covalently bond to the magnetic beads and then incubated with the proteins. However, in the method section, it was stated that the 6-SH probes were used during the cell culture. Which one is the real experimental process?

4.      In Figure 5C, it is obvious that for LUSC patients, high group didn’t have worse prognosis. On the contrary, they had a slightly better OS probability compared to the low group. While the statement in the discussion section was correct, that in result section 3.5 should be changed.

Author Response

We are very grateful to you for giving us an opportunity to revise our manuscript. We want to thank the editors and reviewers for reviewing of manuscript and providing constructive suggestions on our manuscript entitled “6-Shogaol exhibits a promoting effect with Tax via binding HSP60 in non-small cell lung cancer” (Manuscript ID: cells-1994604).

We have studied reviewers’ comments carefully and tried our best to revise our manuscript according to the comments. Thanks again to the hard work! Below are the detailed responses:

Reviewer 2

This study analyzed the potential targets and pathways of 6-Shogaol (6-SH) in NSCLC and demonstrated that 6-SH could be developed as a potential therapeutic agent for NSCLC treatment. The authors demonstrated 6-SH inhibited cancer cell growth both in vitro and in vivo, through induction of apoptosis and cell cycle arrest. Direct target of 6-SH was identified and verified, and other predicted signaling pathways was also validated.

These results have important clinical significance for the treatment of tumors. However, some issues need to be addressed before it could be further considered.

Minor issues:

Comment 1: In Figure 2C, increased nuclear chromatin staining reflects increased chromatin condensation, which is a morphologic hallmark of apoptosis. Please explain more in the result section for the readers who are not that familiar with this feature. Although Hoechst staining was mentioned in the method section, it should still be briefly mentioned either in the main text, in the figure caption, or labeled in Figure 2C.

Response: Thank you for your valuable comments, we have added related description in manuscript (method 2.7, lane 100-101) and marked it in red.

Comment 2:  Figure 3A was not cited in the result section 3.3.

Response: We are very grateful to reviewer for suggesting this question. We have added related description in manuscript (result 3.3, lane 210-212) and marked it in red.

Comment 3: In Figure 4A, it seems with 6-SH probes were first covalently bond to the magnetic beads and then incubated with the proteins. However, in the method section, it was stated that the 6-SH probes were used during the cell culture. Which one is the real experimental process?

Response: We are very grateful to reviewer for suggesting this question. Actually, the technology road-map was omitted the cell culture process. The real experimental process was as described in method section, the 6-SH probes were used during the cell culture before being subjected to SDS-PAGE, Western blotting, and HPLC-mass spectrometry to identify the captured protein target.

Comment 4: In Figure 5C, it is obvious that for LUSC patients, high group didn’t have worse prognosis. On the contrary, they had a slightly better OS probability compared to the low group. While the statement in the discussion section was correct, that in result section 3.5 should be changed.

Response: We accepted the reviewer’s suggestion and changed related description in manuscript (result 3.5, lane 252-254) and marked it in red.

Round 2

Reviewer 1 Report

The menuscript can be accepted in its current form.